# Knowledge and awareness about colorectal cancer and barriers to its screening among a sample of general public in Saudi Arabia

**Muhammad Imran**[1]*, **Mukhtiar Baig**[2], **Razan Obaidallah Alshuaibi**[3], **Thikra Abdullah Almohammadi**[3], **Samah Abdulsalam Albeladi**[3], **Faysal Turki Matuq Zaafarani**[4]

**1** Department of Surgery, Faculty of Medicine in Rabigh, King Abdulaziz University, Jeddah, Saudi Arabia, **2** Department of Clinical Biochemistry, Faculty of Medicine in Rabigh, King Abdulaziz University, Jeddah, Saudi Arabia, **3** 6th Year Medical Student, Faculty of Medicine in Rabigh, King Abdulaziz University, Jeddah, Saudi Arabia, **4** 5th Year Medical Student, Faculty of Medicine in Rabigh, King Abdulaziz University, Jeddah, Saudi Arabia

* minmuhammad@kau.edu.sa

**Data Availability Statement:** All relevant data are within the paper and its Supporting Information files.

## Abstract

### Introduction

The present study investigated knowledge, awareness, and barriers to colorectal cancer (CRC) screening from a sample of the general population in Saudi Arabia.

### Methods

This cross-sectional study was conducted between August 2022 and January 2023 among a sample of the general population in Jeddah, Saudi Arabia. The questionnaire consisted of demographics, knowledge and awareness, and questions about CRC screening barriers.

### Results

A total of 1105 adults belonged to the general public [505 females (45.7%) and 600 males (54.3%)] recruited in this survey. The mean age of the respondents was 39.79±12.49. The internet was the primary source of CC information for most participants, 661(59.8%). Gender-wise comparison of general public responses regarding CRC knowledge and awareness and barriers to screening showed a mixed response. Analysis of participants' knowledge regarding CRC and its risk factors and warning signs showed that 356(32.2%) people believed that the best age for the CRC test is 41–50 years, and 285(25.8%) responded they don't know. Almost half of the participants, 539(48.8%), were not interested in attending awareness seminars about CRC, while 371(33.6%) were interested in attending awareness seminars about CRC. Only one-third of the respondents, 368(33.3%), knew of any tests or examinations used to detect CRC. Participants' perceptions about colonoscopy-related and FOBT-related barriers to CRC screening disclosed that these are time-consuming, expensive, painful, embarrassing, etc.

**Funding:** The author(s) received no specific funding for this work.

**Competing interests:** The authors have declared that no competing interests exist.

## Conclusion

Insufficient information, poor awareness, and several assumed barriers to CRC screening were found among study participants. There is a need to close knowledge gaps and offer them comprehensive information regarding CRC, as well as the availability and benefits of screening. In this aspect, social media can be extremely beneficial.

## Introduction

Colorectal cancer (CRC) is among the top three most prevalent malignancies worldwide, and it is the second cancer leading to death worldwide [1]. CRC is by far the most frequent form of cancer in Saudi Arabia's male population, while in Saudi females it is the third most common cancer. Overall, the incidence rate of CRC has increased in the Saudi population [2]. Though CRC is more common in older patients, it has increased significantly in those under 50 over the past twenty years [3].

Several factors, such as cultural, social, and lifestyle, play a role in the development of CRC [4]. There are multiple risk factors that contribute to CRC, including a positive family history of CRC, people over the age of fifty, a low-fiber or high-fat diet, and the consumption of red meat [5]. In addition, it was found that inflammatory bowel disease increases the risk of CRC [6]. Symptoms of CRC range from altering bowel habits and blood in stool to unintended weight loss [6].

Several years are needed to progress CRC from adenoma into carcinoma. In addition, localized cancer has a 90% survival rate, compared to 10% if metastasis has already occurred [7]. Therefore, screening the population, particularly at-risk ones, will help detect tumors earlier and allow for intervention before cancer develops [7, 8]. Moreover, Colorectal cancer screening (CRCS) methods may reduce the high costs of CRC therapy, such as chemotherapy for patients with late-stage CRC [9].

Screening for CRC is required and recommended for all people over 50. However, people with a positive family history of CRC should start screening at 40 [10]. According to Saudi guidelines, the best screening method for CRC is colonoscopy followed by flexible sigmoidoscopy. In addition, it is recommended to perform a flexible sigmoidoscopy in combination with an annual guaiac occult blood test or fecal immunochemical test every five years if colonoscopy is not an accessible screening option [11].

The ideal screening method should be effective, with high sensitivity and specificity, as well as inexpensive and easily accessible [12]. Nowadays, colonoscopy is the gold standard for CRCS, with a sensitivity of 95%, and it is advised every ten years in adults aged 50 years and over [9]. Furthermore, the removal of precancerous as well as small cancerous cells is one of the most significant benefits of colonoscopy [9]. On the other hand, invasiveness, preparation of the bowel prior to the procedure, and the presence of sedation are limitations of colonoscopy [13]. Despite its availability, it is not considered cheap to the general public, making it a difficult screening method [14]. Flexible sigmoidoscopy is similar to colonoscopy and shares many advantages and limitations [13]. However, there are a few differences between them, as flexible sigmoidoscopy can screen the distal portion of the colon as well as anesthesia and enema are not required [9]. Fecal occult blood test (FOBT) is a frequently used non-invasive procedure to detect CRC. In addition, it is a low-cost method compared to other screening methods [15]. On the other hand, low sensitivity and specificity are considered a disadvantage of FOBT [15].

Some barriers to CRCS have been mentioned, such as a low level of awareness, absence of physicians' advice for screening, fear of screening findings, and absence of CRC symptoms [6]. Additionally, the time required for screening is reported as a barrier to CRCS [16].

Many studies have been conducted worldwide to explore general public knowledge, awareness, and barrier to colorectal cancer (CRC), but the present study investigated knowledge, awareness, and barriers to CRC screening from a sample of the general population in Saudi Arabia. The main objective of this study was to explore knowledge and awareness about colorectal cancer and barriers to its screening among a sample of general public, in a metropolitan city, in Saudi Arabia.

## Methods

The Unit of Biomedical Ethics approved this cross-sectional study, reference No 488–21 (non-Interventional), at the Faculty of Medicine, King Abdulaziz University, Jeddah, Saudi Arabia. Male and female participants of age 18 years or above, Saudi and non-Saudi, were included, while people <18 years were excluded. The general public of Jeddah city was chosen.

The calculated minimum sample size was 400 subjects. The sample size was calculated by the SurveyMonkey sample size calculator [17]. We included a total of 1105 participants to avoid any shortcomings. The study was conducted between August 2022 and January 2023 among the general population in Jeddah, Saudi Arabia. The questionnaire was distributed online via Google Forms. The first page included all the relevant information of research, and consent statement, "I have read all the information provided by the researchers. I understand that my participation is voluntary and that I am free to withdraw at any time, without giving a reason and without cost. I voluntarily agree to take part in this study", was written. The participant could go ahead only in case/he agrees to participate voluntarily. We used a self-administrated questionnaire based on previous similar studies to ensure the questions were valid and reliable [18–22]. The questionnaire consisted of three parts, including demographics, questions about knowledge and awareness, and questions about the CRC screening barriers. Demographics included age, gender, nationality, education level, profession, marital status, monthly income, and source of CRC information. The knowledge part consisted of 22 items. Each item was designed to analyze a particular area of knowledge. The section on barriers to CRC screening consisted of three subsections–general barriers, colonoscopy-related barriers, and fecal occult blood test (FOBT)-related barriers. General barriers included seven items, colonoscopy-related barriers included nine items, and FOBT-related barriers included six items.

The Statistical Package for Social Sciences (SPSS, IBM, USA) software version 26 was used for data entry and analysis. Both descriptive statistics and analytic statistics were examined. Categorical variables were compared by Chi-square test, and the scores were compared by Student's t-test or ANOVA test. The level of statistical significance was set at $p < 0.05$.

## Results

A total of 1105 adults belonged to the general public [505 females (45.7%) and 600 males (54.3%)] recruited in this survey. The mean age of the respondents was 39.79±12.49. Most adults, 892(80.7%), were Saudi nationals. Internet was the primary source of CRC information for most participants, 661(59.8%). Table 1 shows the overall characteristics of the study participants.

Analysis of participants' knowledge regarding colorectal cancer and its risk factors and warning signs showed that 356(32.2%) people believed that the best age for the CRC test is 41–50 years, and 285(25.8%) responded they don't know. Analysis of the question about

**Table 1. General characteristics of the study participants (n = 1105).**

| Variables | Frequency | Percent |
|---|---|---|
| **Age** | | |
| < = 40 | 626 | 56.7 |
| > 40 | 479 | 43.3 |
| **Gender** | | |
| Female | 505 | 45.7 |
| Male | 600 | 54.3 |
| **Nationality** | | |
| Non-Saudi | 213 | 19.3 |
| Saudi | 892 | 80.7 |
| **Education level** | | |
| Bachelor | 613 | 55.5 |
| high school | 250 | 22.6 |
| Illiterate | 12 | 1.1 |
| master | 112 | 10.1 |
| PhD | 37 | 3.3 |
| primary | 81 | 7.3 |
| **Profession** | | |
| business | 213 | 19.3 |
| government | 390 | 35.3 |
| housewife | 197 | 17.8 |
| pvt job | 305 | 27.6 |
| **Marital Status** | | |
| Divorced | 136 | 12.3 |
| Married | 646 | 58.5 |
| Unmarried | 300 | 27.1 |
| Widow | 23 | 2.1 |
| **Monthly income** | | |
| Less than 5000 | 131 | 11.9 |
| 5000–10000 | 396 | 35.9 |
| 10001–15000 | 295 | 26.7 |
| 15001–20000 | 189 | 17.1 |
| More than 20000 | 94 | 8.5 |
| **Source of CRC information** | | |
| Internet | 661 | 59.8 |
| Doctor | 58 | 5.2 |
| Newspaper | 191 | 17.3 |
| None | 195 | 17.6 |

participants' first impression when they heard of CRC, half of the respondents, 557(50.4%), thought it could be cured, and 343(31%) thought it was a fatal disease. One-third of the respondents, 374(33.8%), believed that CRC is commonly found in KSA, while 340(30.8%) responded negatively. Almost half of the participants, 539(48.8%), were not interested in attending awareness seminars about CRC, while 371(33.6%) were interested in attending awareness seminars about CRC. Only one-third of the respondents, 368(33.3%), knew of any tests or examinations used to detect CRC (Table 2).

Most respondents disagreed that CRC is not a severe health threat 662(59.9%) and 167 (15.1%) agreed that CRC screening is ineffective. Participants' perceptions about colonoscopy-

**Table 2. Participants' knowledge regarding colorectal cancer and its risk factors, and warning signs (n = 1105).**

| Questions | Responses | Frequency | Percent |
|---|---|---|---|
| The best age for a test for colon cancer | 20–30 years | 93 | 8.4 |
| | 31–40 years | 141 | 12.8 |
| | 31–40 years | 83 | 7.5 |
| | 41–50 years | 356 | 32.2 |
| | 51–60 years | 147 | 13.3 |
| | don't know | 285 | 25.8 |
| Your first impression when you hear of colon cancer | It's a fatal disease | 343 | 31.0 |
| | it can be cured | 557 | 50.4 |
| | don't know | 205 | 18.6 |
| Commonly affected people by colon cancer | Men | 368 | 33.3 |
| | Women | 341 | 30.9 |
| | don't know | 396 | 35.8 |
| Life expectancy for colon cancer | < 10 yrs | 299 | 27.1 |
| | > 10 yrs | 330 | 29.9 |
| | don't know | 476 | 43.1 |
| Colon cancer is a familial disease | Yes | 220 | 19.9 |
| | No | 433 | 39.2 |
| | don't know | 452 | 40.9 |
| Genetic factors are important for developing CRC disease | Yes | 411 | 37.2 |
| | No | 287 | 26 |
| | don't know | 407 | 36.8 |
| Environmental factors are important for developing CRC disease | Yes | 532 | 48.1 |
| | No | 232 | 21 |
| | don't know | 341 | 30.9 |
| Do you think that colon cancer is commonly found in Saudi Arabia? | Don't know | 391 | 35.4 |
| | No | 340 | 30.8 |
| | Yes | 374 | 33.8 |
| Are you keen to attend awareness seminars about colon cancer? | Don't know | 195 | 17.6 |
| | No | 539 | 48.8 |
| | Yes | 371 | 33.6 |
| Do you think the incidence of colon cancer is influenced by family history? | Don't know | 366 | 33.1 |
| | No | 322 | 29.1 |
| | Yes | 417 | 37.7 |
| Do you think that aging is one of the risk factors for colon cancer? | Don't know | 310 | 28.1 |
| | No | 344 | 31.1 |
| | Yes | 451 | 40.8 |
| Do you think that obesity and lack of exercise are considered risk factors for colon cancer? | Don't know | 298 | 27 |
| | No | 281 | 25.4 |
| | Yes | 526 | 47.6 |
| Do you think that chronic infection of the colon is considered a risk factor for colon cancer? | Don't know | 380 | 34.4 |
| | No | 293 | 26.5 |
| | Yes | 432 | 39.1 |
| Do you think that colon cancer is a preventable disease? | Don't know | 301 | 27.2 |
| | No | 206 | 18.6 |
| | Yes | 598 | 54.1 |

(*Continued*)

**Table 2.** (Continued)

| Questions | Responses | Frequency | Percent |
|---|---|---|---|
| Do you think that colon cancer can be cured if detected at an early stage? | Don't know | 203 | 18.4 |
| | No | 174 | 15.7 |
| | Yes | 728 | 65.9 |
| Have you ever heard about any tests or examination that is used in the detection of colon cancer? | Don't know | 265 | 24 |
| | No | 472 | 42.7 |
| | Yes | 368 | 33.3 |
| Do you think that blood in stool is one of the symptoms related to colon cancer? | Don't know | 380 | 34.4 |
| | No | 271 | 24.5 |
| | Yes | 454 | 41.1 |
| Do you think that fever and weight loss are the symptoms related to colon cancer? | Don't know | 417 | 37.7 |
| | No | 256 | 23.2 |
| | Yes | 432 | 39.1 |
| Do you think that chronic abdominal pain is one of the symptoms related to colon cancer? | Don't know | 99 | 9 |
| | No | 531 | 48 |
| | Yes | 475 | 43 |
| Do you have a family history of colon cancer? | Don't know | 200 | 18 |
| | No | 680 | 61.5 |
| | Yes | 225 | 20.5 |
| I believe CRC screening is not effective | Agree | 167 | 15.1 |
| | Disagree | 547 | 49.5 |
| | Neutral | 391 | 35.4 |
| Colorectal cancer is not a serious health threat | Agree | 157 | 14.2 |
| | Disagree | 662 | 59.9 |
| | Neutral | 286 | 25.9 |

related barriers to CRC screening disclosed that 279(25.2%) agreed with the statement that colonoscopy takes a lot of time, and 162 (14.7%) disagreed. Several participants agreed it is an expensive process 393(35.6%), painful 340(30.8%), embarrassing procedure 433(39.2%), afraid of the colonoscopy results 401(36.3%), fearful of colonoscopy complications 369(33.4%), a previous bad experience with colonoscopy 192(17.4%), and don't know where they can get a colonoscopy 467(42.3%) (Table 3). Participants' perceptions about FOBT-related barriers to CRC screening disclosed that many participants, 244(22.1%), agreed it is an expensive process, 340 (30.8%) that it is a painful procedure, 433(39.2%) embarrassing procedure, 345(31.2%) were afraid of the FOBT results, 230(20.8%) don't have time to get FOBT, 318(28.8%) feeling bad about getting done the FOBT, and 565(51.1%) don't know where they can get an FOBT (Table 3).

Gender-wise comparison of general public responses regarding CC knowledge and awareness are shown in Table 4. Regarding colonoscopy barriers, most of the males, compared to females, believed that colonoscopy screening is not important (p < .001) and they had a previous bad experience with colonoscopy (p = 0.003). More females were afraid of colonoscopy complications compared to males (p = 0.008). Regarding FOBT-related barriers, most females, compared to males, disagreed that FOBT isn't important (p = 0.027) (Table 4).

## Discussion

The current study investigated the general public knowledge and awareness regarding CRC screening and barriers among the Saudi population.

**Table 3. Perceived barriers to CRC screening (n = 1105).**

| Statements | Agree | Neutral | Disagree |
|---|---|---|---|
| **General Barriers** | | | |
| CRC screening is not mandatory | 324(29.3) | 525(47.5) | 256(23.2) |
| I believe CRC screening is not effective | 167(15.1) | 391(35.4) | 547(49.5) |
| Colorectal cancer is not a serious health threat | 157(14.2) | 286(25.9) | 662(59.9) |
| It is difficult to get an appointment with a physician | 354(32) | 414(37.5) | 337(30.5) |
| I don't have transportation | 213(19.3) | 350(31.7) | 542(49) |
| I don't have a physician's recommendation for CRC screening | 545(49.3) | 348(31.5) | 212(19.2) |
| I don't have any symptoms of getting screened for CRC | 591(53.5) | 312(28.2) | 202(18.3) |
| **Colonoscopy related barriers** | | | |
| Colonoscopy takes a lot of time. | 279(25.2) | 556(50.3) | 270(24.4) |
| A colonoscopy isn't important, in my opinion. | 162(14.7) | 418(37.8) | 525(47.5) |
| Colonoscopy is an expensive procedure. | 393(35.6) | 511(46.2) | 201(18.2) |
| I think colonoscopy is very painful. | 340(30.8) | 558(50.5) | 207(18.7) |
| Colonoscopy is a very embarrassing procedure. | 433(39.2) | 404(36.6) | 268(24.3) |
| I am afraid of the results of the colonoscopy. | 401(36.3) | 457(41.4) | 247(22.4) |
| I am afraid of colonoscopy complications. | 369(33.4) | 478(43.3) | 258(23.3) |
| I had a previous bad experience with colonoscopy. | 192(17.4) | 373(33.8) | 540(48.9) |
| I don't know where I can get a colonoscopy. | 467(42.3) | 375(33.9) | 263(23.8) |
| **FOBT related barriers** | | | |
| I think a Fecal occult blood test (FOBT) isn't important. | 204(18.5) | 427(38.6) | 474(42.9) |
| FOBT is an expensive procedure | 244(22.1) | 578(52.3) | 283(25.6) |
| I don't have time to get a test for FOBT. | 230(20.8) | 445(40.3) | 430(38.9) |
| I am afraid of the results of the FOBT. | 345(31.2) | 453(41) | 307(27.8) |
| I am feeling bad about getting done the FOBT. | 318(28.8) | 462(41.8) | 325(29.4) |
| I don't know where I can get the FOBT. | 565(51.1) | 307(27.8) | 233(21.1) |

We found that the general community has insufficient knowledge of the warning signs and risk factors of colorectal cancer. Most respondents from the target population received their education regarding CRC screening online. This study showed that less than half of the participants believed that aging, family history, chronic colon infection, and a lack of activity were risk factors for CRC. One-third of the participants identified the main warning signs as blood in stool, chronic abdominal pain, fever, and weight loss. This study aligned with a substantial body of evidence pointing toward a low knowledge of CRC in many countries. A study was done in the Riyadh region, where knowledge and awareness of CRC were low [23]. Another survey on CRC knowledge among the Omani adult population reported insufficient knowledge of CRC [22]. This finding suggests that individuals are not knowledgeable enough about cancer risk factors, indicating a need for awareness programs about CRC. It is anticipated that their lack of knowledge may further reduce their chances of screening and early cancer detection.

Furthermore, although 65.9% of the participants knew that colon cancer could be cured if detected at an early stage and 59.9% had knowledge that it is a serious disease; however, approximately almost half of the participants indicated that they had never heard about any tests or examinations that are used in the detection of colon cancer. This study's results were similar to a Chinese study [24]. The present study results showed that poor awareness of screening tests is a factor contributing to the low participation rate of CRC screening and indicates that screening tests could be a good target for cancer awareness-raising initiatives.

**Table 4. Gender-wise comparison of reported barriers to CRC screening (n = 1105).**

| General Barriers | Responses | Female (n = 505) | Male (600) | P-value |
|---|---|---|---|---|
| CRC screening is not mandatory | Agree | 140(27.7) | 184(30.7) | 0.561 |
| | Disagree | 119(23.6) | 137(22.8) | |
| | Neutral | 246(48.7) | 279(46.5) | |
| I believe CRC screening is not effective | Agree | 63(12.5) | 104(17.3) | <0.001 |
| | Disagree | 289(57.2) | 258(43) | |
| | Neutral | 153(30.3) | 238(39.7) | |
| Colorectal cancer is not a serious health threat | Agree | 66(13.1) | 91(15.2) | 0.040 |
| | Disagree | 323(64) | 339(56.5) | |
| | Neutral | 116(23) | 170(28.3) | |
| It is difficult to get an appointment with a physician | Agree | 133(26.3) | 221(36.8) | <0.001 |
| | Disagree | 183(36.2) | 154(25.7) | |
| | Neutral | 189(37.4) | 225(37.5) | |
| I don't have transportation | Agree | 105(20.8) | 108(18) | 0.484 |
| | Disagree | 24147.7% | 301(50.2) | |
| | Neutral | 159(31.5) | 191(31.8) | |
| I don't have a physician's recommendation for CRC screening | Agree | 243(48.1) | 302(50.3) | 0.756 |
| | Disagree | 100(19.8) | 112(18.7) | |
| | Neutral | 162(32.1) | 186(31) | |
| I don't have any symptoms of getting screened for CRC | Agree | 294(58.2) | 297(49.5) | 0.013 |
| | Disagree | 80(15.8) | 122(20.3) | |
| | Neutral | 131(25.9) | 181(30.2) | |
| **Colonoscopy related barriers** | | | | |
| Colonoscopy takes a lot of time | Agree | 120(23.8) | 159(26.5) | 0.520 |
| | Disagree | 129(25.5) | 141(23.5) | |
| | Neutral | 256(50) | 300(50) | |
| A colonoscopy isn't important, in my opinion | Agree | 51(10.1) | 111(18.5) | <0.001 |
| | Disagree | 267(52.9) | 258(43) | |
| | Neutral | 187(37) | 231(38.5) | |
| Colonoscopy is an expensive procedure | Agree | 190(37.6) | 203(33.8) | 0.399 |
| | Disagree | 91(18) | 110(18.3) | |
| | Neutral | 224(44.4) | 287(47.8) | |
| I think colonoscopy is very painful | Agree | 189(37.4) | 151(25.2) | 0.541 |
| | Disagree | 88(17.4) | 119(19.8) | |
| | Neutral | 228(45.1) | 330(55) | |
| Colonoscopy is a very embarrassing procedure | Agree | 209(41.4) | 224(37.3) | 0.173 |
| | Disagree | 110(21.8) | 158(26.3) | |
| | Neutral | 186(36.8) | 218(36.3) | |
| I am afraid of the results of the colonoscopy | Agree | 196(38.8) | 205(34.2) | 0.240 |
| | Disagree | 105(20.8) | 142(23.7) | |
| | Neutral | 204(40.4) | 253(42.2) | |
| I am afraid of colonoscopy complications | Agree | 193(38.2) | 176(29.3) | 0.008 |
| | Disagree | 110(21.8) | 148(24.7) | |
| | Neutral | 202(40) | 276(46) | |
| I had a previous bad experience with colonoscopy | Agree | 70(13.9) | 122(20.3) | 0.003 |
| | Disagree | 272(53.9) | 268(44.7) | |
| | Neutral | 163(32.3) | 210(35) | |

(*Continued*)

**Table 4.** (Continued)

| General Barriers | Responses | Female (n = 505) | Male (600) | P-value |
|---|---|---|---|---|
| I don't know where I can get a colonoscopy | Agree | 210(41.6) | 257(42.8) | 0.627 |
| | Disagree | 127(25.1) | 136(22.7) | |
| | Neutral | 168(33.3) | 207(34.5) | |
| **FOBT related Barriers** | | | | |
| I think a Fecal occult blood test (FOBT) isn't important | Agree | 96(19) | 108(18) | 0.027 |
| | Disagree | 235(46.5) | 239(39.8) | |
| | Neutral | 174(34.5) | 253(42.2) | |
| FOBT is an expensive procedure | Agree | 109(21.6) | 135(22.5) | 0.487 |
| | Disagree | 138(27.3) | 145(24.2) | |
| | Neutral | 258(51.1) | 320(53.3) | |
| I don't have time to get a test for FOBT | Agree | 112(22.2) | 118(19.7) | 0.476 |
| | Disagree | 198(39.2) | 232(38.7) | |
| | Neutral | 195(38.6) | 250(41.7) | |
| I am afraid of the results of the FOBT | Agree | 155(30.7) | 190(31.7) | 0.517 |
| | Disagree | 134(26.5) | 173(28.8) | |
| | Neutral | 216(42.8) | 237(39.5) | |
| I am feeling bad about getting done the FOBT | Agree | 153(30.3) | 165(27.5) | 0.581 |
| | Disagree | 144(28.5) | 181(30.2) | |
| | Neutral | 208(41.2) | 254(42.3) | |
| I don't know where I can get the FOBT | Agree | 261(51.7) | 304(50.7) | 0.873 |
| | Disagree | 103(20.4) | 130(21.7) | |
| | Neutral | 141(27.9) | 166(27.7) | |

This study revealed a variety of barriers to CRC screening in the Saudi population. A lack of general public awareness that CRC screening is limited for persons with symptoms was the most common barrier; the result is similar to a study by Yong et al. [25]. Numerous studies have reported the lack of physicians' recommendations as one of the most common barriers. According to a recent study from Saudi Arabia, the most common barrier to not being screened for CRC is the lack of physician recommendations [6]. This barrier was the second most common in our research. The fear of CRC screening results is one of the common obstacles observed in another study [26], which we reported in our study as well.

In our study, we found that 22.1% think that the expensive cost of FOBT is a barrier, while 52.3% were neutral, and 25.6% disagree that FOBT is expensive, which roughly agrees with another research finding [25]. Most of the results found in our research about FOBT barriers were neutral. We predict that they have bad knowledge and awareness about "what FOBT is" and its importance because research found that there is no awareness about FOBT. They discovered that just 24.8% of people were aware of colonoscopy or FOBT as a CRC screening technique [25].

According to the findings of this study, there is a critical need to increase knowledge, awareness, and barriers to CRC screening among the general Saudi population. The current study's findings can help policymakers target at-risk individuals by filling gaps in knowledge, approaches, and barriers linked to CRC, a threatening health problem in the Kingdom. Future initiatives to improve ordinary people's knowledge and fill knowledge gaps should focus on effective mass campaigns using digital and social media channels. Educating them about CRC screening, its availability, and its utility is necessary.

The current study has significant limitations. First, the cause-and-effect relationship cannot be measured because this is a cross-sectional study. Many participants' responses in questionnaire-based studies do not reflect their genuine responses; instead, they tick the most relevant answer. As a result, selection bias cannot be avoided in such investigations. Because the data does not accurately represent all population of Saudi Arabia, the conclusions cannot be generalized. However, it provides a good picture of knowledge, awareness, and barriers to CRC screening in a sample of the general Saudi population in the Jeddah region.

## Conclusions

The current study revealed insufficient information, poor awareness, and several assumed barriers to CRC screening. There is a need to close knowledge gaps and offer them comprehensive information regarding the CRC and the availability and benefits of screening. In this aspect, social media can be extremely beneficial.

## Supporting information

**S1 File.**
(SAV)

## Author Contributions

**Conceptualization:** Muhammad Imran.

**Data curation:** Muhammad Imran, Mukhtiar Baig.

**Formal analysis:** Mukhtiar Baig.

**Investigation:** Muhammad Imran, Mukhtiar Baig, Razan Obaidallah Alshuaibi, Thikra Abdullah Almohammadi, Samah Abdulsalam Albeladi, Faysal Turki Matuq Zaafarani.

**Methodology:** Muhammad Imran.

**Project administration:** Muhammad Imran.

**Resources:** Razan Obaidallah Alshuaibi, Thikra Abdullah Almohammadi, Samah Abdulsalam Albeladi, Faysal Turki Matuq Zaafarani.

**Validation:** Muhammad Imran, Mukhtiar Baig, Razan Obaidallah Alshuaibi, Thikra Abdullah Almohammadi, Samah Abdulsalam Albeladi, Faysal Turki Matuq Zaafarani.

**Visualization:** Muhammad Imran, Mukhtiar Baig, Razan Obaidallah Alshuaibi, Thikra Abdullah Almohammadi, Samah Abdulsalam Albeladi, Faysal Turki Matuq Zaafarani.

**Writing – original draft:** Muhammad Imran, Mukhtiar Baig.

**Writing – review & editing:** Muhammad Imran, Mukhtiar Baig, Razan Obaidallah Alshuaibi, Thikra Abdullah Almohammadi, Samah Abdulsalam Albeladi, Faysal Turki Matuq Zaafarani.

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
