## [Decision Letter · Decision Letter 0]

17 May 2023

PONE-D-23-08312Knowledge and awareness about colorectal cancer and barriers to its screening among a sample of general public in Saudi ArabiaPLOS ONE

Dear Dr. Imran,

Thank you for submitting your manuscript to PLOS ONE. After careful consideration, we feel that it has merit but does not fully meet PLOS ONE’s publication criteria as it currently stands. Therefore, we invite you to submit a revised version of the manuscript that addresses the points raised during the review process.

The reviewers have provided some very useful suggestions and I would like you to address all of them. In addition, you must expand your Method section substantially. For example, you must provide information about the sampling procedure, where the participants came from, what were the dates of assessment and so forth. You must provide information about your measures. Were they all closed questions or were some open ended (as I suspect)? With respect to the Results, you must explain what analyses were conducted and you cannot just provide p values. If you have run analyses on means, you should not provide the whole distribution. Please organize these into a meaningful, separate table. Do not use terms like "Illiterate". You seem to mean, "No Formal Education", or something similar. Please revise your table 2 and make sure to be consistent. The order of the response options (Y/N/DK) flips in this table.  There must be consistency with the formats of Tables 2 and 3. Table 2 should be more like Table 3. Strongly consider supplementing your presentation with further significance testing, where appropriate. This will substantially increase the usefulness of this manuscript to researchers. I must emphasise that your treatment of these issues is critical. The manuscript is not publishable as it is, but if you substantially improve these sections, I am willing to consider the revision. I strongly suggest you consult with an experienced colleague and/or check a recognized style guide, such as the APA Style Manual 7th ed, or equivalent. These sources will provide a very clear guide to what is necessary in each manuscript section.

We look forward to receiving your revised manuscript.

Kind regards,

Stefano Occhipinti

Academic Editor

PLOS ONE

Journal Requirements:

Reviewers' comments:

Reviewer's Responses to Questions

**Comments to the Author**

1. Is the manuscript technically sound, and do the data support the conclusions?

Reviewer #1: Yes

Reviewer #2: Partly

2. Has the statistical analysis been performed appropriately and rigorously? 

Reviewer #1: Yes

Reviewer #2: No

3. Have the authors made all data underlying the findings in their manuscript fully available?

Reviewer #1: Yes

Reviewer #2: Yes

4. Is the manuscript presented in an intelligible fashion and written in standard English?

Reviewer #1: Yes

Reviewer #2: Yes

5. Review Comments to the Author

Reviewer #1: The topic is very interesting. Relevant in connection with the increase in oncological morbidity in the world. Screening programs are a good tool for detecting cancer at an early stage. Generally, it is well-written and concise.

Questions for the author:

1. You indicated that you used a self-administrated questionnaire. The questionnaire validation procedure is not described in detail by the author, but this is not critical. There are standard techniques that the author of this article probably applied.

2. What method did the author use to collect data in this study? In the Methods section, the authors should indicate the research design. Were the respondents randomly selected? Random selection of respondents is an important part of the cross-sectional study. The procedure for distributing the questionnaire to respondents and collecting information should be described in more detail.

Reviewer #2: The authors have provided descriptive findings of an important area of research, relating to CRC screening knowledge, awareness and barriers. Limitations of the current manuscript relate to the detail of findings and the mostly descriptive nature of the conducted analyses. Further analysis and development of rationale for the study would improve this manuscript. Further detail is provided for the specific manuscript sections below.

- Abstract

o more detail needed regarding gender-wise comparison findings

- Introduction

o clarity needed for discussion of Saudi screening guidelines

o limited detail regarding existing literature

o more detail regarding existing literature is required for barriers identified

o increase rationale for study – why is the Saudi population likely to differ from previous studies and the gap in literature

o no rationale for gender-wise comparisons provided

- Method

o How were participants recruited?

o More detail regarding measures needed – validated measures? Developed for this study? If so, based on what?

o The introduction discusses flexible sigmoidoscopy as a Saudi guideline for screening, however no measures assess flexible sigmoidoscopy knowledge, awareness or barriers

- Results

o Results are presented in repetition with table findings, and provided in a purely descriptive manner

o FOBT barriers do not address common barriers to FOBT relating to the screening modality – e.g., faecal collection

o Gender-wise comparisons only completed for barriers and not knowledge/awareness

o Given age-related factors for screening and CRC, analysis should be conducted to assess age differences in knowledge and barriers

- Discussion

o Greater detail regarding previous literature needed, as opposed to e.g., stating similar to a Chinese study

o Claim that “The present study results showed that poor awareness of screening tests is a factor contributing to the low participation rate of CRC screening and indicates that screening tests could be a good target for cancer awareness-raising initiatives.” is unfounded in this study, as the study did not investigate screening participation via either intention or behaviour

o Attempt made to integrate findings with previous literature, however this is presented as superficial statements regarding the same finding

o Claim regarding at-risk individuals may not be accurate given the sample included people over the age of 18 and no attempt to analyse age differences was made

o Limitations stated regarding questionnaire and genuine responses should be referenced, and further speak to the need to address this with the study method chosen (i.e., social desirability measures, attention checks, etc.)

o Link between questionnaire and selection bias made, however this is not presented accurately with selection bias relevant to study sample and recruitment and not participant responding

- General

o Review of typographical errors and written language

6. PLOS authors have the option to publish the peer review history of their article (what does this mean?). If published, this will include your full peer review and any attached files.

Reviewer #1: No

Reviewer #2: No

---

## [Author Response · Author response to Decision Letter 0]

18 Jul 2023

PONE-D-23-08312

Knowledge and awareness about colorectal cancer and barriers to its screening among a sample of general public in Saudi Arabia

PLOS ONE

Dear academic editor, 

We appreciate your thorough feedback. We value all of your's and the reviewers' comments and recommendations. We have taken practically all of your suggestions into consideration. After adopting all of the comments, we believe the manuscript's quality has greatly improved. 

Thank you for your time and extremely constructive feedback.

With best regards,

Dr Muhammad Imran

Corresponding author.

Editor’s comments 

The reviewers have provided some very useful suggestions, and I would like you to address all of them. In addition, you must expand your Method section substantially. For example, you must provide information about the sampling procedure, where the participants came from, the dates of assessment, and so forth. You must provide information about your measures. Were they all closed questions, or were some open-ended (as I suspect)?

 Response: The methods section is expanded as suggested. We used all closed-ended questions. 

With respect to the results, you must explain what analyses were conducted and you cannot just provide p values. If you have run analyses on means, you should not provide the whole distribution. Please organize these into a meaningful, separate table. 

Response: Thank you for pointing out this important issue. We applied the Chi-square test in table 4. We have added this to the table. We have reanalyzed our data, and three more tables have been constructed as suggested by the reviewers.

Do not use terms like "Illiterate". You seem to mean, "No Formal Education", or something similar. 

Response: Modified as suggested. 

Please revise your table 2 and make sure to be consistent. The order of the response options (Y/N/DK) flips in this table. There must be consistency with the formats of Tables 2 and 3. Table 2 should be more like Table 3. 

Response: Table 2 has been completely modified as Table 3. Now there is consistency in the repones and format.

Table 4 has also been modified.

Strongly consider supplementing your presentation with further significance testing, where appropriate. This will substantially increase the usefulness of this manuscript to researchers.

Response: In Table 4, there was a gender-wise comparison, so we used the chi-square test, while in Tables 2 and 3, there are frequencies and percentages against each statement. Therefore, the comparison test is not applicable. 

We have further analyzed our data and included three more tables.

I must emphasize that your treatment of these issues is critical. The manuscript is not publishable as it is, but if you substantially improve these sections, I am willing to consider the revision. I strongly suggest you consult with an experienced colleague and/or check a recognized style guide, such as the APA Style Manual 7th ed or equivalent. These sources will provide a very clear guide to what is necessary in each manuscript section.

Response:

The manuscript is revised according to your suggestions and the suggestions of the reviewers. A senior colleague has reviewed the manuscript. 

Response: The manuscript has been structured according to PLOS ONE’s style requirements. 

4. Please provide additional details regarding participant consent. In the ethics statement in the Methods and online submission information, please ensure that you have specified (1) whether consent was informed and (2) what type you obtained (for instance, written or verbal, and if verbal, how it was documented and witnessed). If your study included minors, state whether you obtained consent from parents or guardians. If the need for consent was waived by the ethics committee, please include this information.

Response: Thanks for pointing this out. Informed consent was taken, and it is included in the methods section along with the procedure. The study did not include minors. 

Reviewers' comments:

5. Review Comments to the Author

Reviewer #1: The topic is very interesting. Relevant in connection with the increase in oncological morbidity in the world. Screening programs are a good tool for detecting cancer at an early stage. Generally, it is well-written and concise.

Questions for the author:

1. You indicated that you used a self-administrated questionnaire. The questionnaire validation procedure is not described in detail by the author, but this is not critical. There are standard techniques that the author of this article probably applied.

Response: The questionnaire validation procedure was applied. It is now included in the methods section. 

2. What method did the author use to collect data in this study? In the Methods section, the authors should indicate the research design. Were the respondents randomly selected? Random selection of respondents is an important part of the cross-sectional study. The procedure for distributing the questionnaire to respondents and collecting information should be described in more detail.

Response: This cross-sectional study's method is now mentioned in more detail in the methods section. Yes, you are right randomization is an important concern. As it was an online questionnaire and we did not have complete access to the population of Jeddah city, we could not do simple randomization or cluster method; instead, we used the snowball technique, which is now mentioned in the methods section. This point is also mentioned as a limitation. 

Reviewer #2: The authors have provided descriptive findings of an important area of research relating to CRC screening knowledge, awareness, and barriers. Limitations of the current manuscript relate to the detail of findings and the mostly descriptive nature of the conducted analyses. Further analysis and development of the rationale for the study would improve this manuscript. Further detail is provided for the specific manuscript sections below.

- Abstract

o more detail needed regarding gender-wise comparison findings

Response: More details have been added. 

- Introduction

o clarity needed for discussion of Saudi screening guidelines

o limited detail regarding existing literature

o more detail regarding existing literature is required for barriers identified

o increase rationale for study – why is the Saudi population likely to differ from previous studies and the gap in literature

o no rationale for gender-wise comparisons provided

Response: The suggested points are included in the introduction. 

- Method

o How were participants recruited?

o More detail regarding measures needed – validated measures? Developed for this study? If so, based on what?

o The introduction discusses flexible sigmoidoscopy as a Saudi guideline for screening, however no measures assess flexible sigmoidoscopy knowledge, awareness or barriers

Response: The details of participants’ recruitment, details regarding validated measures (for the questionnaire), and other details are added in the methods section. Thanks for pointing out an important issue of including questions about flexible sigmoidoscopy in the questionnaire. We admit it is very important, but it is our mistake that we didn't include this. One reason was to take questions from the published studies. Unfortunately, it cannot be done now. Therefore, we are including this point in the limitations.

- Results

o Results are presented in repetition with table findings, and provided in a purely descriptive manner

Response: Description of the results have been modified.

o FOBT barriers do not address common barriers to FOBT relating to the screening modality – e.g., faecal collection

Response: Thanks for indicating this important issue, which is missing. We are including in the limitation section. 

o Gender-wise comparisons only completed for barriers and not knowledge/awareness

Response: A new table has been constructed showing gender-wise comparisons of Participants' knowledge regarding colorectal cancer and its risk factors and warning signs (Table 3).

o Given age-related factors for screening and CRC, analysis should be conducted to assess age differences in knowledge and barriers.

Response: A new table has been constructed showing age-wise comparison of participants’ knowledge regarding colorectal cancer and its risk factors, and warning signs (Table 4).

A new table has been constructed showing the age-wise comparison of barriers regarding colorectal cancer (Table 7).

- Discussion

o Greater detail regarding previous literature needed, as opposed to e.g., stating similar to a Chinese study

Response: More detail is added. 

o Claim that "The present study results showed that poor awareness of screening tests is a factor contributing to the low participation rate of CRC screening and indicates that screening tests could be a good target for cancer awareness-raising initiatives." is unfounded in this study, as the study did not investigate screening participation via either intention or behaviour

Response: Thanks for pointing out. The sentences are removed. 

o Attempt made to integrate findings with previous literature; however, this is presented as superficial statements regarding the same finding

Response: Comparison is added in the discussion section. 

o Claim regarding at-risk individuals may not be accurate given the sample included people over the age of 18 and no attempt to analyse age differences was made

Response: A new table has been constructed showing age-wise comparison of participants' knowledge regarding colorectal cancer and its risk factors and warning signs (Table 4).

A new table has been constructed showing the age-wise comparison of barriers regarding colorectal cancer (Table 7).

The sentence is modified accordingly. 

o Limitations stated regarding questionnaire and genuine responses should be referenced, and further speak to the need to address this with the study method chosen (i.e., social desirability measures, attention checks, etc.)

Response: We have rephrased the sentence. A sentence attention checks in this questionnaire and the presence of social desirability bias have been included in the methods section. A reference has also been included. 

o Link between questionnaire and selection bias made, however this is not presented accurately with selection bias relevant to study sample and recruitment and not participant responding. Response: Response: We have rephrased the sentence.

- General

o Review of typographical errors and written language

Response: throughout the manuscript, language editing has been done.

---

## [Editor Report · Decision Letter 1]

4 Aug 2023

Knowledge and awareness about colorectal cancer and barriers to its screening among a sample of general public in Saudi Arabia

PONE-D-23-08312R1

Dear Dr. Imran,

We’re pleased to inform you that your manuscript has been judged scientifically suitable for publication and will be formally accepted for publication once it meets all outstanding technical requirements.

Kind regards,

Stefano Occhipinti

Academic Editor

PLOS ONE
---

## [Editor Report · Acceptance letter]

14 Aug 2023

PONE-D-23-08312R1 

Knowledge and awareness about colorectal cancer and barriers to its screening among a sample of general public in Saudi Arabia 

Dear Dr. Imran:

I'm pleased to inform you that your manuscript has been deemed suitable for publication in PLOS ONE. Congratulations! Your manuscript is now with our production department. 

Kind regards, 

on behalf of

Prof. Stefano Occhipinti 

Academic Editor

PLOS ONE